# Impact of Inflammatory Burden on Voriconazole Exposure in Oncohematological Pediatric Patients Receiving Antifungal Prophylaxis after Allogeneic HCT

**DOI:** 10.3390/microorganisms12071388

**Published:** 2024-07-09

**Authors:** Milo Gatti, Caterina Campoli, Edoardo Muratore, Tamara Belotti, Riccardo Masetti, Marcello Lanari, Pierluigi Viale, Federico Pea

**Affiliations:** 1Department of Medical and Surgical Sciences, Alma Mater Studiorum University of Bologna, 40138 Bologna, Italy; riccardo.masetti5@unibo.it (R.M.); marcello.lanari@unibo.it (M.L.); pierluigi.viale@unibo.it (P.V.); federico.pea@unibo.it (F.P.); 2Clinical Pharmacology Unit, Department for Integrated Infectious Risk Management, IRCCS Azienda Ospedaliero-Universitaria di Bologna, 40138 Bologna, Italy; 3Infectious Diseases Unit, Department for Integrated Infectious Risk Management, IRCCS Azienda Ospedaliero-Universitaria di Bologna, 40138 Bologna, Italy; caterina.campoli@aosp.bo.it; 4Pediatric Hematology and Oncology, IRCCS Azienda Ospedaliero-Universitaria di Bologna, 40138 Bologna, Italy; edoardo.muratore@studio.unibo.it (E.M.); tamara.belotti@aosp.bo.it (T.B.); 5Pediatric Emergency Unit, IRCCS Azienda Ospedaliero-Universitaria di Bologna, 40138 Bologna, Italy

**Keywords:** oncohematologic paediatric patients, allogeneic hematopoietic stem cell transplantation, voriconazole, primary antifungal prophylaxis, inflammation, interleukin-6, C-reactive protein, procalcitonin

## Abstract

(1) Background: The impact of inflammation on voriconazole exposure in oncohematological pediatric patients represents a debated issue. We aimed to investigate the impact of serum C-reactive protein (CRP), procalcitonin (PCT), and interleukin-6 (IL-6) levels on voriconazole exposure in oncohematological pediatric patients requiring allogeneic hematopoietic stem cell transplantation (HCT). (2) Methods: Pediatric patients undergoing allogeneic HCT and receiving therapeutic drug monitoring (TDM)-guided voriconazole as primary antifungal prophylaxis between January 2021 and December 2023 were included. The ratio between concentration and dose (C/D) of voriconazole was used as a surrogate marker of total clearance. A receiving operating characteristic curve analysis was performed by using CRP, PCT, or IL-6 values as the test variable and voriconazole C/D ratio > 0.188 or >0.375 (corresponding to a trough concentration value [C_min_] of 3 mg/L normalized to the maintenance dose of 16 mg/kg/day in patients of age < 12 years and of 8 mg/kg/day in those ≥12 years, respectively) as the state variable. Area under the curve (AUC) and 95% confidence interval (CI) were calculated. (3) Results: Overall, 39 patients were included. The median (IQR) voriconazole C_min_ was 1.7 (0.7–3.0) mg/L. A CRP value > 8.49 mg/dL (AUC = 0.72; 95%CI 0.68–0.76; *p* < 0.0001), a PCT value > 2.6 ng/mL (AUC = 0.71; 95%CI 0.63–0.77; *p* < 0.0001), and an IL-6 value > 27.9 pg/mL (AUC = 0.80; 95%CI 0.71–0.88; *p* < 0.0001) were significantly associated with voriconazole overexposure. Consistent results were found in patients aged <12 and ≥12 years. (4) Conclusions: A single specific threshold of inflammatory biomarkers may be linked to a significantly higher risk of voriconazole exposure in oncohematological pediatric patients after HCT, irrespective of age. Adopting a TDM-guided strategy could be useful for minimizing the risk of voriconazole overexposure.

## 1. Introduction

Invasive fungal infections (IFIs) may represent a major cause of morbidity and mortality among pediatric patients requiring allogeneic hematopoietic stem cell transplantation (HCT) [1,2]. Implementing primary antifungal prophylaxis is mandatory in this scenario because of the concomitance of several risk factors, namely the occurrence of severe and persisting neutropenia and/or mucositis, the use of corticosteroids, and/or the positioning of an indwelling central venous catheter [1,2,3,4].

Voriconazole is currently a first-line agent for prophylaxis and treatment of IFIs in oncohematologic children [1,2]. Voriconazole pharmacokinetics (PK) may be affected by wide intra- and inter-individual variability [5,6], so that therapeutic drug monitoring (TDM) of trough concentrations (C_min_) is recommended for personalizing the dosing regimen, granting optimal exposure, and minimizing the risk of toxicity [7,8].

Among the several factors affecting voriconazole PK [5,6], inflammation was recently associated with voriconazole overexposure, leading to potential toxicity risk [6,9,10,11,12,13,14]. Elevated levels of some pro-inflammatory cytokines, namely IL1-β, IL-6, and TNF-α, were shown to downregulate to a variable extent the activity of CYP2C9, CYP2C19, and CYP3A4,6,7, namely the three CYP450 isoenzymes metabolizing voriconazole. Particularly, interleukin 6 (IL-6) was shown to downregulate strongly CYP3A4, moderately CYP2C19, and mildly CYP2C9 [5,6,14,15]. Additionally, it may be expected that inflammatory biomarkers like CRP and procalcitonin (PCT) that are triggered by sepsis and/or by the cytokine burden [16,17] might also have an impact on voriconazole metabolism.

Elevated levels of circulating pro-inflammatory cytokines, especially IL-6, can be triggered by several underlying diseases and may lead to a life-threatening systemic inflammatory syndrome [18]. Among oncohematologic pediatric patients undergoing HCT, cytokine release syndrome may be triggered by pre-engraftment or engraftment syndrome, mucositis-related tissue damage, acute graft-versus-host disease [GvHD], and/or sepsis [19]. Accordingly, monitoring serum levels of IL-6 and of surrogate inflammatory biomarkers like CRP and PCT is extremely important in the early post-HCT period. 

How the inflammation may impact voriconazole exposure was only partially assessed in the setting of oncohematologic pediatric patients receiving voriconazole treatment in the early post-HCT period. CRP levels were shown to affect voriconazole exposure in patients aged ≥12 years but not in those aged <12 years [9,20,21]. However, no study has yet assessed the impact of IL-6 and/or PCT levels on voriconazole exposure.

Based on these assumptions, the aim of this study was to investigate the impact that serum levels of CRP, PCT, and IL-6 may have on voriconazole exposure in oncohematological pediatric patients requiring allogeneic HCT.

## 2. Materials and Methods

### 2.1. Study Design

This retrospective study was carried out among oncohematological patients aged <21 years who were admitted to the Pediatric Oncohaematology Transplant Unit of the IRCCS Azienda Ospedaliero–Universitaria of Bologna, Italy, in the period 1 January 2021–31 December 2023 and received primary antifungal prophylaxis with voriconazole after undergoing allogeneic HCT for any indication. Patients were included if they underwent at least one simultaneous assessment of both voriconazole C_min_ and one or more inflammatory biomarkers among CRP, PCT, and/or IL-6 levels. The study was approved by the Ethics Committee of IRCCS Azienda Ospedaliero-Universitaria of Bologna (n. 442/2021/Oss/AOUBo approved on 28 June 2021).

### 2.2. Data Collection

Demographic (age, sex, height, weight, body surface area [BSA], underlying hematological diseases, donor type, and stem cell source) and clinical/laboratory data (voriconazole dosing regimen and treatment duration, concomitant medications acting as modulators of CYP 2C9, CYP2C19, and CYP3A4 activity, serum CRP, PCT, and IL-6 levels, serum galactomannan levels, occurrence of breakthrough IFI, and occurrence of HCT complications [i.e., acute and/or chronic GvHD, documented Gram-negative bloodstream infections during febrile neutropenia, need for ICU admission]) were retrieved for each patient. Strict monitoring of serum inflammatory biomarkers, namely CRP, PCT, and IL-6, was adopted as routine clinical practice in the early post-HCT period. Serum CRP levels were determined by means of an immunoturbidimetric method (normal value < 0.5 mg/dL). Serum PCT levels were determined by means of an enzyme-linked immunosorbent assay (normal value < 0.1 ng/mL). Serum IL-6 levels were determined by means of an electrochemiluminescence immunoassay (normal value < 7.8 pg/mL). Specifically, serum CRP levels were monitored every 24–48 h throughout the hospital stay, and serum PCT levels were measured twice weekly and whenever febrile neutropenia occurred, whereas serum IL-6 levels were monitored in the first three weeks after HCT at the onset of each febrile neutropenia episode and in the first day of defervescence. Breakthrough IFI was defined according to predefined international criteria [22]. A serum galactomannan value ≥ 1 was defined as a significant threshold for probable invasive aspergillosis [22].

### 2.3. Voriconazole Dosing Regimens, Sampling Procedure, and Definition of Optimal Exposure 

The initial voriconazole dosing regimen was based on the summary of product recommendations. In patients aged 2–11 years or in those aged 12–14 years weighing < 50 kg, an intravenous (IV) loading dose (LD) of 9 mg/kg every 12 h for the first 24 h was followed by a maintenance dose (MD) of 8 mg/kg every 12. In patients aged ≥12 years weighing ≥ 50 kg, an IV LD of 6 mg/kg every 12 h in the first 24 h was followed by an MD of 4 mg/kg every 12 h. In patients aged <2 years, since there is a lack of a summary of product recommendations, the same approach was adopted as was used for patients aged 2–11 years, based on what was previously suggested in the literature [8].

Voriconazole dosing adjustments were performed according to a real-time TDM-based expert clinical pharmacological advice program, as previously reported [23]. As per routine clinical practice, blood samples for determining plasma voriconazole C_min_ were collected 5–15 min before administering the drug after at least 48 h from starting treatment and further reassessed every 48–72 h for promptly identifying voriconazole under- or overexposure. Voriconazole plasma concentrations were measured by means of a liquid chromatography–tandem mass spectrometry (LC–MS/MS) commercially available method (Chromsystems Instruments and Chemicals GmbH, Munich, Germany) [24]. The selected voriconazole C_min_ range was defined at 1.0–3.0 mg/L according to recent meta-analyses showing that voriconazole C_min_ > 3.0 mg/L was associated with significantly higher risk of hepatotoxicity and neurotoxicity. Consequently, voriconazole C_min_ > 3.0 mg/L was considered voriconazole overexposure and defined as a potentially toxic level [25,26,27].

### 2.4. Statistical Analysis

To allow an accurate comparison of TDM data coming from the different age classes of our patient population, voriconazole C_min_ data were normalized per kg of body weight by dividing them by the daily voriconazole dose and were expressed as C_min_/D ratio in mg/L per mg/kg/daily. Subsequently, the voriconazole C_min_/D ratio threshold of toxicity for each subgroup was calculated by inserting in this formula the voriconazole potentially toxic level (namely 3 mg/L) as C_min_ and the voriconazole MD adopted in each subgroup as D (namely, 16 mg/kg/daily in pediatric patients aged ≥12 years or 8 mg/kg/daily in those aged <12 years or aged 12–14 years and weighing < 50 kg). The corresponding voriconazole C_min_/D ratio thresholds of toxicity resulted in >0.188 and >0.375, respectively. Receiving operating characteristic (ROC) curve analysis was performed to assess the potential impact of the inflammatory burden on voriconazole overexposure by inserting the serum CRP, PCT, or IL-6 values as the test variables and the voriconazole C_min_/D ratio thresholds of toxicity as the state variable. The area under the curve (AUC), along with a 95% confidence interval (CI), was calculated. The optimal cut-off point was retrieved according to the Youden Index method. The Youden Index was calculated by means of the following equation: sensitivity (%) + specificity (%) − 100. Overall analysis and subgroup analysis based on age classes (namely <12 years and ≥12 years) were performed by calculating the AUC for each of the selected inflammatory biomarkers. We decided to carry out the age classes subgroup analysis because of the conflicting findings observed previously in these two age groups concerning the impact of CRP levels on voriconazole exposure [9,20,21]. Statistical analysis was performed by using MedCalc for Windows (MedCalc statistical software, version 19.6.1, MedCalc Software Ltd., Ostend, Belgium). *p* values < 0.05 were considered statistically significant. Continuous data were presented as median and interquartile range (IQR), whereas categorical variables were expressed as count and percentage. 

## 3. Results

Overall, a total of 39 pediatric patients undergoing allogeneic HCT received a TDM-guided primary antifungal prophylaxis with voriconazole (Table 1). 

The median (IQR) age was 10 (5–15) years, with a male preponderance (66.7%). Twenty-one patients (53.8%) were aged 2–11 years and one <2 years. Acute lymphoblastic leukemia (51.2%) and acute myeloid leukemia (25.6%) represented the most frequent underlying hematological diseases requiring allogeneic HCT. 

The median (IQR) duration of voriconazole prophylaxis was 46 (33.5–75) days. The median (IQR) number of TDM assessments of voriconazole C_min_ per patient was 13 (8.5–21.5), and the median number of days (IQR) to first TDM assessment was 3 (2.5–4). The median (IQR) average voriconazole C_min_ was 1.7 mg/L (0.7–3.2 mg/L) in patients aged <12 years, with a median (IQR) average voriconazole daily dose of 18 mg/kg (13.1–26.4 mg/kg). In patients aged ≥12 years, the median (IQR) average voriconazole C_min_ was 1.4 mg/L (0.8–2.8 mg/L), with a median (IQR) average voriconazole daily dose of 9.1 mg/kg (5.9–12.1 mg/kg). 

All of the included patients were cotreated with CYP2C9, CYP2C19, or CYP3A4 modulators. The most frequent were proton pump inhibitors (97.4%), corticosteroids (66.7%), and letermovir (15.4%).

No patient experienced any breakthrough IFI while on voriconazole prophylaxis. The serum galactomannan index was always negative, with a median (IQR) value of 0.11 (0.07–0.16). Four patients (10.3%) experienced suspected voriconazole toxicity (two hepatotoxicity with an increase in serum transaminases, and one each neurotoxicity and skin rash) and had antifungal prophylaxis switched to liposomal amphotericin B (in three cases) or to isavuconazole (in the other one). Fourteen patients (35.9%) had febrile neutropenia coupled with documented Gram-negative bacteremia (nine required intensive care unit admission [23.1%]), and twenty (51.3%) experienced acute or chronic GvHD). 

A comparison of clinical features in patients aged <12 and ≥12 years is provided in Table 2. Patients aged ≥12 years had significantly higher rates of both documented Gram-negative bacteremia (58.9% vs. 18.2%; *p* = 0.017) and ICU admission (41.2% vs. 9.1%; *p* = 0.026) and a trend toward higher letermovir coadministration (29.4% vs. 4.5%; *p* = 0.06) compared to those aged <12 years. 

A comparative analysis of serum inflammatory biomarker levels between cases having voriconazole overexposure and those having normal and/or under-exposure is provided in Appendix A. The causes and specific time onset of inflammations are detailed in Appendix A. The ROC analysis results are summarized in Table 3. 

Regarding the overall group of patients, first, a total of 599 paired voriconazole C_min_-CRP assessments were included, and ROC curve analysis found that a CRP value > 8.49 mg/dL was significantly associated with voriconazole overexposure (sensitivity of 53.2% and specificity of 85.6%), with an AUC of 0.72 (95%CI 0.68–0.76; *p* < 0.0001; Figure 1). 

Second, a total of 247 paired voriconazole C_min_-PCT assessments were included, and ROC curve analysis found that a PCT value > 2.6 ng/mL was significantly linked to voriconazole overexposure (sensitivity of 53.2% and specificity of 86.9%), with an AUC of 0.71 (95%CI 0.63–0.77; *p* < 0.0001; Figure 2). 

Third, a total of 93 paired voriconazole C_min_-IL-6 were included, and ROC curve analysis showed that an IL-6 value > 27.9 pg/mL was significantly associated with voriconazole overexposure (sensitivity of 97.5% and specificity of 50.9%), with an AUC of 0.80 (95%CI 0.71–0.88; *p* < 0.0001; Figure 3). 

In regard to the subgroup analysis based on age classes, the ROC curve analysis confirmed that voriconazole overexposure was significantly associated with threshold values of all three inflammatory biomarkers, both in patients aged <12 years [CRP > 5.49 mg/dL; AUC 0.68; 95%CI 0.63–0.71; *p* < 0.0001. PCT > 2.92 ng/mL; AUC 0.63; 95%CI 0.54–0.71; *p* = 0.01. IL-6 > 27.9 pg/mL; AUC 0.76; 95%CI 0.64–0.86; *p* < 0.0001] and in those aged ≥12 years [CRP > 12.38 mg/dL; AUC 0.92; 95%CI 0.87–0.95; *p* < 0.0001. PCT > 2.4 ng/mL; AUC 0.86; 95%CI 0.78–0.91; *p* < 0.0001. IL-6 > 52.0 pg/mL; AUC 0.87; 95%CI 0.71–0.97; *p* < 0.0001] (Table 3).

## 4. Discussion

To the best of our knowledge, this is the first study that has investigated the impact of the inflammatory burden based on CRP, PCT, and IL-6 serum levels on voriconazole exposure among oncohematological pediatric patients after HCT. Our findings suggest that in both the age classes, specific serum thresholds for each of these inflammatory biomarkers may be significantly linked to an increase in the likelihood of causing voriconazole overexposure, which, in turn, could increase the risk of drug-related hepato- and/or neurotoxicity [25,26,27].

Some previous studies assessed the potential impact of inflammation on voriconazole exposure among pediatric patients [20,21,28,29]. However, unlike ours, all of these tested only the impact of one biomarker, namely CRP, and none of these performed ROC curve analysis to identify potential significant thresholds associated with voriconazole overexposure. Overall, all of these studies suggested that a high CRP burden may be associated with voriconazole overexposure, especially among children aged >10 years. A retrospective study carried out among 27 children showed that a categorical arbitrary threshold of CRP > 150 mg/L was associated with higher voriconazole C_min_ among patients aged ≥12 years (5.8 vs. 2.2 mg/L; *p* = 0.027; *n* = 16) but not among those aged <12 years (3.3 vs. 2.6 mg/L, *p* = 0.682; *n* = 11) [20]. Likewise, another retrospective study including 52 hematological pediatric patients after HCT found that a categorical arbitrary threshold of CRP > 40 mg/L tended toward a significantly higher voriconazole C_min_ among patients aged 11–18 years (*p* = 0.08; *n* = 21) but not among those aged 2–10 years (*p* = 0.60; *n* = 31) [21]. A prospective observational study including 27 oncohematologic pediatric patients aged 2–12 years (of which 66.7% were undergoing HCT) found that a categorical arbitrary threshold of CRP > 40 mg/L was associated with high voriconazole C_min_ > 5.5 mg/L (*p* = 0.03) [29]. Conversely, a retrospective study including 61 pediatric patients (mean age 10.3 years; >75% affected by hematological disease, 50% of whom were undergoing HCT) found by multivariate analysis that a CRP level < 110 mg/L was an independent predictor of low voriconazole C_min_ < 1.0 mg/L (*p* = 0.045) [28].

Our findings first suggest that among oncohematological pediatric patients receiving voriconazole prophylaxis after HCT, the impact of inflammation on voriconazole exposure may be significantly linked at the ROC analysis to well-defined thresholds for each of three major inflammatory biomarkers, namely CRP, PCT, and IL-6. Subgroup analysis by age showed that the absolute value of the thresholds of CRP and IL-6 linked to voriconazole overexposure was higher in patients aged ≥12 years than in those aged <12 years (CRP: 12.38 vs. 5.49 mg/dL; IL-6: 52.0 vs. 27.9 pg/mL). This finding seems to be at odds with the fact that the baseline expression and catalytic efficiency of CYP2C19, namely the major isoform metabolizing voriconazole, is expected to be higher in pediatric patients compared with adults [30]. Indeed, this discrepancy could be at least partially explained by the trend toward more frequent letermovir coadministration among patients aged ≥12 years, namely a potent inducer of CYP2C19 that might have upregulated its expression [31,32].

Interestingly, ROC analysis showed that IL-6 was the most sensitive in predicting voriconazole overexposure among the three inflammatory biomarkers tested. This is in agreement with the findings of several preclinical and clinical studies showing that IL-6 strongly downregulates CYP3A4, moderately CYP2C19, and mildly CYP2C9,17,18, thus decreasing voriconazole metabolism. It is worth mentioning that the IL-6 threshold identified in our analysis (27.9 pg/mL) is of similar extent to that (18 pg/mL) identified in a previous study carried out among COVID-19 adult patients who were associated with impaired CYP3A4-mediated darunavir clearance after CART analysis [33].

The majority of oncohematologic pediatric patients receiving voriconazole prophylaxis at standard doses experienced voriconazole overexposure on one or more occasions during the first three weeks after HCT. This highlights once more the mandatory role that a real-time TDM-based expert clinical pharmacological advice program of voriconazole should have in this setting for allowing prompt personalization of antifungal prophylaxis in hematological children, as previously reported in other settings [13,34,35,36].

Interestingly, in clinical settings where the real-time TDM-based approach is unavailable or unfeasible, the identified thresholds of serum inflammatory biomarker levels could also represent a helpful tool for clinicians in adopting voriconazole dosing decreases or in considering shifting therapy to other antifungals. In this latter case, it should not be overlooked that other issues may arise, namely the risk of nephrotoxicity with liposomal amphotericin B [37] and of hepatotoxicity with posaconazole or isavuconazole. Additionally, clinically relevant drug–drug interactions may be expected with both posaconazole and isavuconazole [38,39], and an impact of inflammation on isavuconazole metabolism could not be ruled out, considering that it is a substrate of CYP3A4 [5].

The limitations of our study should be addressed. The retrospective monocentric design and the limited number of included patients must be recognized. The potential impact of either CYP2C19 genetic polymorphism or drug interactions could not be ruled out. Conversely, assessing the impact of multiple inflammatory biomarkers by ROC analysis may represent a point of strength.

In conclusion, our findings suggest that in oncohematological pediatric patients receiving voriconazole prophylaxis after HCT, single specific thresholds of IL-6, CRP, and PCT serum values may be linked to a significantly higher risk of voriconazole overexposure, irrespective of age. Adopting a TDM-guided strategy and assessing the inflammatory status in the early post-HCT period may play an important role in minimizing the risk of voriconazole overexposure, potentially leading to drug-related toxicity. Large prospective studies are warranted to confirm our findings.

## Figures and Tables

**Figure 1 microorganisms-12-01388-f001:**
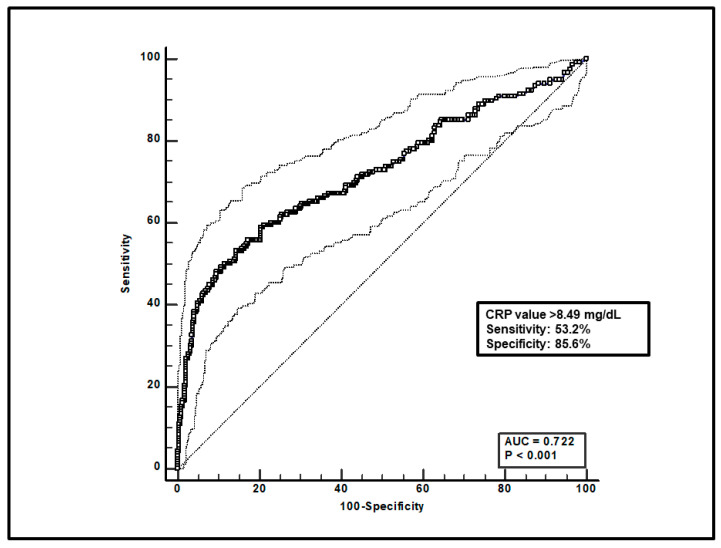
ROC curve analysis for voriconazole C_min_ > 3 mg/L. The 100-specificity (false-positive rate) and sensitivity (true-positive rate) are plotted on the X and Y axes, respectively. An optimal cut-off of CRP value > 8.49 mg/dL was found, with a sensitivity of 53.2% and specificity of 85.6%. Continuous and dotted lines represent the ROC curve and 95% confidence intervals, respectively.

**Figure 2 microorganisms-12-01388-f002:**
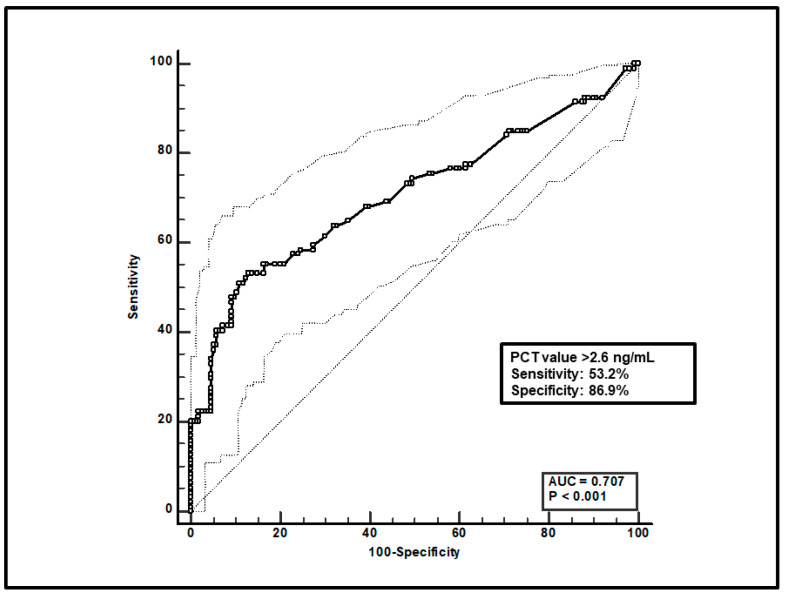
ROC curve analysis for voriconazole C_min_ > 3 mg/L. The 100-specificity (false-positive rate) and sensitivity (true-positive rate) are plotted on the X and Y axes, respectively. An optimal cut-off of PCT value > 2.6 ng/mL was found, with a sensitivity of 53.2% and specificity of 86.9%. Continuous and dotted lines represent the ROC curve and 95% confidence intervals, respectively.

**Figure 3 microorganisms-12-01388-f003:**
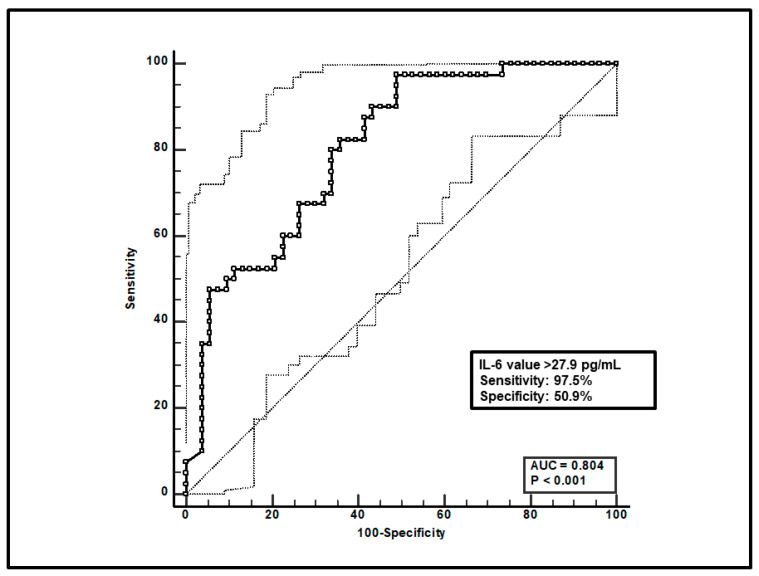
ROC curve analysis for voriconazole C_min_ > 3 mg/L. The 100-specificity (false-positive rate) and sensitivity (true-positive rate) are plotted on the X and Y axes, respectively. An optimal cut-off of IL-6 value > 27.9 pg/mL was found, with a sensitivity of 97.5% and specificity of 50.9%. Continuous and dotted lines represent the ROC curve and 95% confidence intervals, respectively.

**Table 1 microorganisms-12-01388-t001:** Demographics and clinical characteristics of hematologic pediatric patients undergoing allogeneic HCT and receiving voriconazole as primary antifungal prophylaxis.

Patient Demographic	Patients (N = 39)
Age (years) [median (IQR)]	10 (5–15)
Age < 2 years	1 (2.6)
Age 2–11 years	21 (53.8)
Age ≥ 12 years	17 (43.6)
Gender (male/female) [*n* (%)]	26/13 (66.7–33.3)
Body weight (kg) [median (IQR)]	41.0 (19.7–61.0)
Body surface area (m^2^) [median (IQR)]	1.29 (0.78–1.67)
Underlying oncohematologic disease [*n* (%)]	
ALL	20 (51.2)
AML	10 (25.6)
JMML	2 (5.1)
Fanconi anemia	1 (2.6)
Beta-thalassemia	1 (2.6)
Aplastic anemia	1 (2.6)
Anaplastic large cell lymphoma	1 (2.6)
HL	1 (2.6)
NHL	1 (2.6)
Myelodysplastic syndrome	1 (2.6)
Donor [*n* (%)]	
MUD	21 (53.9)
Haploidentical	16 (41.0)
Sibling	2 (5.1)
Stem cell source [*n* (%)]	
Bone Marrow	26 (66.7)
Peripheral Blood	13 (33.3)
Voriconazole prophylaxis	
Median dose (mg/kg/daily) [median (IQR)]	13.9 (8.8–22.2)
Length of prophylaxis [days; median (IQR)]	46 (33.5–75.0)
No. of TDM assessments per patient [median (IQR)]	13 (8.5–21.5)
Average C_min_ (mg/L) [median (IQR)]	1.7 (0.7–3.0)
Median time to first TDM (days) [median (IQR)]	3 (2.5–4)
No. of patients experiencing voriconazole overexposure during the first three weeks after HCT	31 (79.5)
Serum inflammatory biomarkers level [median (IQR)]	
C-reactive protein (mg/dL)	1.38 (0.41–8.07)
Procalcitonin (ng/mL)	0.8 (0.3–3.0)
Interleukin-6 (pg/mL)	72.8 (21.4–270.6)
Concomitant agents [*n* (%)]	
Modulators of CYP2C9, CYP2C19 and/or CYP3A4	39 (100.0)
Proton pump inhibitors	38 (97.4)
Corticosteroids	26 (66.7)
Letermovir	6 (15.4)
HCT complications [*n* (%)]	
Documented Gram-negative bacteremia	14 (35.9)
ICU admission	9 (23.1)
Acute and/or chronic GvHD	20 (51.3)
Clinical outcome [*n* (%)]	
Breakthrough IFI	0 (0.0)
Voriconazole withdrawal due to suspected toxicity	4 (10.3)

ALL: acute lymphoblastic leukemia; AML: acute myeloid leukemia; C_min_: trough concentrations; GvHD: graft-versus-host disease; HL: Hodgkin lymphoma; HCT: hematopoietic stem cell transplant; ICU: intensive care unit; IFI: invasive fungal infection; IQR: interquartile range; JMML: juvenile myelomonocytic leukemia; MUD: Matched Unrelated Donor; NHL: non-Hodgkin lymphoma; TDM: therapeutic drug monitoring. Data are presented as median (IQR) for continuous variables and as *n* (%) for dichotomous variables.

**Table 2 microorganisms-12-01388-t002:** Comparison of clinical features between hematologic pediatric patients <12 and ≥12 years undergoing allogeneic HCT and receiving voriconazole as primary antifungal prophylaxis.

Clinical Variables	Patients < 12 Years (N = 22)	Patients ≥ 12 Years (N = 17)	*p* Value
Underlying oncohematologic disease [*n* (%)]			
ALL	10 (45.5)	10 (58.8)	0.41
AML	7 (31.9)	3 (17.6)	0.46
JMML	2 (9.1)	0 (0.0)	0.50
Fanconi anemia	1 (4.5)	0 (0.0)	0.99
Beta-thalassemia	1 (4.5)	0 (0.0)	0.99
Aplastic anemia	0 (0.0)	1 (5.9)	0.44
Anaplastic large cell lymphoma	1 (4.5)	0 (0.0)	0.99
HL	0 (0.0)	1 (5.9)	0.44
NHL	0 (0.0)	1 (5.9)	0.44
Myelodysplastic syndrome	0 (0.0)	1 (5.9)	0.44
Voriconazole prophylaxis			
Median dose (mg/kg/daily) [median (IQR)]	18.0 (13.1–26.4)	9.1 (5.9–12.1)	<0.0001
Length of prophylaxis [days; median (IQR)]	56.5 (42.75–107.75)	38 (26–48)	0.005
No. of TDM assessments per patient [median (IQR)]	14 (9.25–24.25)	12 (7–14)	0.19
Average C_min_ (mg/L) [median (IQR)]	1.7 (0.7–3.2)	1.4 (0.8–2.8)	0.43
Median time to first TDM (days) [median (IQR)]	4 (3–5)	3 (2–3)	0.002
No. of patients experiencing voriconazole overexposure in the first three weeks after HCT [*n* (%)]	20 (90.9)	11 (64.7)	0.06
Concomitant agents			
Modulators of CYP2C9, 2C19, and/or 3A4	22 (100.0)	17 (100.0)	0.42
Proton pump inhibitors	22 (100.0)	16 (94.1)	0.44
Corticosteroids	17 (77.3)	9 (52.9)	0.11
Letermovir	1 (4.5)	5 (29.4)	0.06
HCT complications			
Documented Gram-negative bacteremia	4 (18.2)	10 (58.9)	0.017
ICU admission	2 (9.1)	7 (41.2)	0.026
Acute and/or chronic GVHD	13 (59.1)	7 (41.2)	0.34
Clinical outcome [*n* (%)]			
Breakthrough IFI	0 (0.0)	0 (0.0)	0.99
Voriconazole withdrawn for suspected toxicity	4 (18.2)	0 (0.0)	0.12

ALL: acute lymphoblastic leukemia; AML: acute myeloid leukemia; GVHD: graft-versus-host disease; HL: Hodgkin lymphoma; HCT: hematopoietic stem cell transplant; ICU: intensive care unit; IFI: invasive fungal infection; IQR: interquartile range; JMML: juvenile myelomonocytic leukemia; NHL: non-Hodgkin lymphoma. Data are presented as *n* (%).

**Table 3 microorganisms-12-01388-t003:** ROC analysis identifying specific inflammatory biomarker thresholds for increased risk of voriconazole overexposure (C_min_ > 3 mg/L) stratified by age.

Inflammatory Biomarkers	No. of Assessments	Serum Threshold Value	AUC ROC Curve (95%CI)	Sensitivity	Specificity	*p* Value
Overall						
C-reactive protein	599	8.49 mg/dL	0.72 (0.68–0.76)	53.2%	85.6%	<0.0001
Procalcitonin	247	2.6 ng/mL	0.71 (0.63–0.77)	53.2%	86.9%	<0.0001
IL-6	93	27.9 pg/mL	0.80 (0.71–0.88)	97.5%	50.9%	<0.0001
Age < 12 years						
C-reactive protein	390	5.49 mg/dL	0.68 (0.63–0.71)	46.0%	85.6%	<0.0001
Procalcitonin	132	2.92 ng/mL	0.63 (0.54–0.71)	33.9%	93.4%	0.01
IL-6	61	27.9 pg/mL	0.76 (0.64–0.86)	96.3%	52.9%	<0.0001
Age ≥ 12 years						
C-reactive protein	209	12.38 mg/dL	0.92 (0.87–0.95)	83.7%	89.2%	<0.0001
Procalcitonin	115	2.4 ng/mL	0.86 (0.78–0.91)	81.6%	81.8%	<0.0001
IL-6	32	52.0 pg/mL	0.87 (0.71–0.97)	92.3%	73.7%	<0.0001

AUC: area under curve; CI: confidence interval; IL-6; interleukin-6; ROC: receiving operating characteristic.

## Data Availability

The data presented in this study are available on request from the corresponding author. The data are not publicly available due to privacy concerns.

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
