# Peer review of "Impact of Inflammatory Burden on Voriconazole Exposure in Oncohematological Pediatric Patients Receiving Antifungal Prophylaxis after Allogeneic HCT"

_microorganisms, 2024, doi:10.3390/microorganisms12071388_

Round 1

Reviewer 1 Report

Comments and Suggestions for Authors

Dear authors

Thank you very much for this article. It's a very nice article with exciting results.

Do you have any data about the patient being treated with isavuconazole? Have you observed the same results as with voriconazole?

Kind regards

Author Response

RESPONSE TO REVIEWERS

Manuscript ID: microorganisms-3092454 entitled “Impact of inflammatory burden on voriconazole exposure in oncohematological pediatric patients receiving antifungal prophylaxis after allogenic hematopoietic stem cell transplantation” by Gatti et al.

Dear Editor,

We would like to thank you for the opportunity to resubmit a revised version of this manuscript. We appreciated the reviewers’ constructive comments. All have been carefully considered and incorporated, where and whenever possible, in the revision. Furthermore, as suggested by the Editorial Office, we rewrote the highlighted parts as “red 1” in which some overlaps were detected.

Our point-by-point responses are provided below.

Q= QUERY; A= ANSWER

Reviewer #1

Q1. Thank you very much for this article. It's a very nice article with exciting results. Do you have any data about the patient being treated with isavuconazole? Have you observed the same results as with voriconazole?

A1. We thank the reviewer for appreciating our manuscript. Considering that the administration of isavuconazole is currently off-label in pediatric patients, our available data in this scenario are very limited, and do not allow for performing a similar analysis to that with voriconazole. However, according to the fact that isavuconazole is primarily metabolized by CYP3A4, it could be expected that a similar pharmacokinetic behaviour could also occur with isavuconazole in the presence of inflammation. As also suggested by Reviewer #3, we discussed this topic in the Discussion section (refer to Line 287-295).

Reviewer 2 Report

Comments and Suggestions for Authors

The manuscript of Milo Gatti and coauthors is devoted to the study of impact of inflammation on voriconazole exposure in oncohematological pediatric patients. The article as a whole is written in a structured and understandable manner. However, a number of clarifications should be made to the text of the manuscript.

General comments

-The title is too long and does not reflect the essence of the study. The authors did not study the effect of inflammation on voriconazole concentrations in the blood, but compared the presence of inflammation and voriconazole overexposure.

-It is not clear from the Abstract and Introduction the need to control the concentration of voriconazole. How common are toxic effects of voriconazole?

-Some details (concentrations, percentages, reliability) should be removed from the abstract to make it easier to understand in my opinion.

-It has been written about how IL-6 can affect voriconazole levels in the blood. But what about C-reactive protein and procalcitonin? Is there a direct connection in this case or are these simply conditions under which voriconazole may be cleared from the body more slowly? How exactly an increase in these parameters can be associated with voriconazole overexposure (at least an assumption).

-Why was the dose of voriconazole for children >12 years old lower than for younger children?

-The Materials and Methods section should be divided on subsection. It is worth indicating methods for determining IL-6, CRP and PCT as well as their reference values. It is not clear from the description what the curve represents (Figure 1-3), what is plotted along the X and Y axes, what sensitivity and specificity are.

-“Patients were included if underwent at least one simultaneous assessment of both voriconazole Cmin and one or more inflammatory biomarkers among CRP, PCT, and/or IL-6 levels”. These patient inclusion criteria indicate that this was not a targeted study. Was the selection of patients for analysis based on the availability of these data?

-Based on Table 1 and information from the text, it follows that the concentrations of voriconazole in patients were not exceeded (C min 0.7-3.0, not >3 “Consequently, voriconazole Cmin >3.0 mg/L was considered voriconazole overexposure and defined as potentially toxic level”)? But “No. of patients experiencing voriconazole overexposure during the first three weeks after HSCT was 31 patients”. What does it mean?

-Why are data about voriconazole concentration not shown in Table 2 for different ages of children?

-I think it would be necessary to provide separately, perhaps in supporting materials, the measured parameters specifically for patients with an overexposure of voriconazole. In addition, comparison of all three parameters (CRP, PCT and IL-6) would be useful in such patients

- How justified is it to present only average parameters without analyzing the specific situation of patients? For example, comparisons of possible causes of inflammation. I think a more detailed analysis of specific data would allow us to draw more conclusions. In any case, data on all patients should be provided in supporting materials.

-How exactly can the obtained data be useful? Will the presence of severe inflammation in a patient be a signal for the doctor to prescribe another antifungal drug or reduce the administered dose of voriconazole? What about the toxicity of those drugs that are prescribed instead of voriconazole. Amphotericin B, as far as I know, is even more toxic than voriconazole. What about the effect of inflammation on the pharmacokinetics of other antifungal drugs?

Minor comments

- Tables contain empty columns. Line 119 needs correction.

Author Response

RESPONSE TO REVIEWERS

Manuscript ID: microorganisms-3092454 entitled “Impact of inflammatory burden on voriconazole exposure in oncohematological pediatric patients receiving antifungal prophylaxis after allogenic hematopoietic stem cell transplantation” by Gatti et al.

Dear Editor,

We would like to thank you for the opportunity to resubmit a revised version of this manuscript. We appreciated the reviewers’ constructive comments. All have been carefully considered and incorporated, where and whenever possible, in the revision. Furthermore, as suggested by the Editorial Office, we rewrote the highlighted parts as “red 1” in which some overlaps were detected.

Our point-by-point responses are provided below.

Q= QUERY; A= ANSWER

Reviewer #2

The manuscript of Milo Gatti and coauthors is devoted to the study of impact of inflammation on voriconazole exposure in oncohematological pediatric patients. The article as a whole is written in a structured and understandable manner. However, a number of clarifications should be made to the text of the manuscript.

General comments

Q1. The title is too long and does not reflect the essence of the study. The authors did not study the effect of inflammation on voriconazole concentrations in the blood, but compared the presence of inflammation and voriconazole overexposure.

A1. We thank the reviewer for this suggestion, and we shortened the title as required. However, the aim of our study was to assess the impact of inflammation burden on voriconazole exposure by identifying potential significant thresholds of inflammatory biomarkers, and not to compare the presence of inflammation and voriconazole overexposure. Consequently, we think that the selected title could be eligible for describing the study.

Q2. It is not clear from the Abstract and Introduction the need to control the concentration of voriconazole. How common are toxic effects of voriconazole?

A2. We thank the reviewer for this comment. We added in the Introduction section (Line 54) the importance of monitoring voriconazole trough concentrations for granting optimal exposure and minimizing the risk of toxicity. Indeed, different meta-analyses (refer to references 25-27) showed that voriconazole trough concentrations < 3.0 mg/L were associated with 63% and 48% significantly lower risk of hepatotoxicity and neurotoxicity, respectively. Consequently, adopting a TDM-guided approach for maximizing could be useful for minimizing voriconazole overexposure and consequent risk of toxicity.

Q3. Some details (concentrations, percentages, reliability) should be removed from the abstract to make it easier to understand in my opinion.

A3. Thank you for this suggestion. However, the current version of the abstract has 250 words, being Results section of 61 words and limited only to essential information. We think that reporting concentrations and percentages is essential for providing a summary of the main findings of the study.

Q4. It has been written about how IL-6 can affect voriconazole levels in the blood. But what about C-reactive protein and procalcitonin? Is there a direct connection in this case or are these simply conditions under which voriconazole may be cleared from the body more slowly? How exactly an increase in these parameters can be associated with voriconazole overexposure (at least an assumption).

A4. We thank the reviewer for this comment, allowing us to better clarify this issue. It should be noted that a direct connection exists between IL-6 and CRP serum levels, considering that following initial synthesis, IL‑6 is transported via the bloodstream to the liver, where it stimulates synthesis of the inflammatory acute phase proteins including CRP. Similarly, it should be noted that infection represents one of the most important causes of inflammation, being strictly associated with raise in serum PCT and CRP levels. Consequently, a direct relationship may be retrieved between the three different inflammatory biomarkers and voriconazole metabolism. We detailed this issue in the Introduction section (refer to Line 61-63).

Q5. Why was the dose of voriconazole for children >12 years old lower than for younger children?

A5. We thank the reviewer for this comment, allowing us to better clarify this issue. As reported in the Materials and Methods and Discussion section (Line 112-119 and 267-270), the dose of voriconazole in pediatric patients ≥ 12 years is lower as per summary of product recommendations according to a higher baseline expression and catalytic efficiency of CYP2C19 in pediatric patients < 12 years.

Q6. The Materials and Methods section should be divided on subsection. It is worth indicating methods for determining IL-6, CRP and PCT as well as their reference values. It is not clear from the description what the curve represents (Figure 1-3), what is plotted along the X and Y axes, what sensitivity and specificity are.

A6.  Thank you for this suggestion. We divided the Materials and Methods section in different subsections, and we added the methods and reference values of the different inflammatory biomarkers (refer to Line 100-104). Furthermore, we detailed Figure 1-3 legends in order to improve clarity.  

Q7. “Patients were included if underwent at least one simultaneous assessment of both voriconazole Cmin and one or more inflammatory biomarkers among CRP, PCT, and/or IL-6 levels”. These patient inclusion criteria indicate that this was not a targeted study. Was the selection of patients for analysis based on the availability of these data?

A7. We thank the reviewer for this comment, allowing us to better clarify this relevant issue. As reported in the Materials and Methods section (refer to Line 82-90), our study has a retrospective design, thus not representing a targeted study, as specific among limitations (refer to Line 296-298). However, it should be noted that according to routine clinical practice adopted at our center, all pediatric patients receiving HCT and treated with voriconazole prophylaxis underwent a TDM-guided approach consisting in the first assessment of trough concentrations at 48-72 hours followed by eventual dosing adjustment and further reassessments every 48-72 hours in order to promptly identifying voriconazole under- or overexposure, as detailed in Materials and Methods section (refer to Line 122-125). Furthermore, as per routine clinical practice, a careful assessment of serum inflammatory biomarkers was performed at different specific timepoints in these patients, as detailed in Materials and Methods section (refer to Line 104-108), thus data were available for all pediatric HCT performed in the study period.

Q8. Based on Table 1 and information from the text, it follows that the concentrations of voriconazole in patients were not exceeded (C min 0.7-3.0, not >3 “Consequently, voriconazole Cmin >3.0 mg/L was considered voriconazole overexposure and defined as potentially toxic level”)? But “No. of patients experiencing voriconazole overexposure during the first three weeks after HSCT was 31 patients”. What does it mean?

A8. We thank the reviewer for this comment, allowing us to better clarify this issue. As reported in Table 1, the median average voriconazole Cmin was 1.7 mg/L, with an interquartile range of 0.7-3.0 mg/L. This means that 25% of voriconazole Cmin was above 3.0 mg/L, and consequently classified as overexposure. This justified the fact that 31 out of 39 included patients (79.5%) experienced at least one voriconazole Cmin above 3.0 mg/L in the first three weeks after HCT. We specified in Table 1 that average voriconazole Cmin were indicated as median and IQR.

Q9. Why are data about voriconazole concentration not shown in Table 2 for different ages of children?

A9. We thank the reviewer for this suggestion. We added data on voriconazole concentrations according to different ages in Table 2.

Q10. I think it would be necessary to provide separately, perhaps in supporting materials, the measured parameters specifically for patients with an overexposure of voriconazole. In addition, comparison of all three parameters (CRP, PCT and IL-6) would be useful in such patients

A10. We thank the reviewer for this suggestion. We provided a comparison of serum inflammatory biomarkers values between episodes of voriconazole overexposure and those in therapeutic range and/or underexposure in Supplementary Table 1.

Q11. How justified is it to present only average parameters without analyzing the specific situation of patients? For example, comparisons of possible causes of inflammation. I think a more detailed analysis of specific data would allow us to draw more conclusions. In any case, data on all patients should be provided in supporting materials.

A11. We thank the reviewer for this comment. We added a specific figure (Supplementary Figure 1) in which causes of inflammation and timeline are detailed for each included patient.

Q12. How exactly can the obtained data be useful? Will the presence of severe inflammation in a patient be a signal for the doctor to prescribe another antifungal drug or reduce the administered dose of voriconazole? What about the toxicity of those drugs that are prescribed instead of voriconazole. Amphotericin B, as far as I know, is even more toxic than voriconazole. What about the effect of inflammation on the pharmacokinetics of other antifungal drugs?

A12. We thank the reviewer for this comment, allowing us to better detail these relevant aspects concerning the clinical application of our findings, as well as the toxicity and the impact of inflammatory burden on pharmacokinetic behaviour of other antifungal agents in the Discussion section (refer to Line 287-295).

Minor comments

Q13. Tables contain empty columns. Line 119 needs correction.

A13. Thank you for this suggestion. We corrected the typo at Line 134, and we removed empty columns from the tables.

Reviewer 3 Report

Comments and Suggestions for Authors

In the paper entitled "Impact of inflammatory burden on voriconazole exposure in oncohematological pediatric patients receiving antifungal prophylaxis after allogenic hematopoietic stem cell transplantation", the authors investigated  the impact of serum CRP, PCT and IL-6 on voriconazole exposure in oncohematological pediatric patients requiring allogenic HCT.

As expected and as this study proves, inflammation may indeed play a pivotal role in voriconazole or other drug exposure.  Certainly, the medical field needs many more of this kind of studies. 

Regarding the manuscript, it is well organized and very well written. It only needs a minor revision before acceptance:

- I was wondering the term "pediatric" could be really accepted considering that the study also included patients between 18 and 20 (<21)...that are basically proper adults;

- page 3 line 107: replace "12h" with "12 h" (space);

- page 6 line 153: delete the comma between twenty and one (correct: twenty one);

- maybe the quality of the figures could be improved.

Author Response

RESPONSE TO REVIEWERS

Manuscript ID: microorganisms-3092454 entitled “Impact of inflammatory burden on voriconazole exposure in oncohematological pediatric patients receiving antifungal prophylaxis after allogenic hematopoietic stem cell transplantation” by Gatti et al.

Dear Editor,

We would like to thank you for the opportunity to resubmit a revised version of this manuscript. We appreciated the reviewers’ constructive comments. All have been carefully considered and incorporated, where and whenever possible, in the revision. Furthermore, as suggested by the Editorial Office, we rewrote the highlighted parts as “red 1” in which some overlaps were detected.

Our point-by-point responses are provided below.

Q= QUERY; A= ANSWER

Reviewer #3

In the paper entitled "Impact of inflammatory burden on voriconazole exposure in oncohematological pediatric patients receiving antifungal prophylaxis after allogenic hematopoietic stem cell transplantation", the authors investigated  the impact of serum CRP, PCT and IL-6 on voriconazole exposure in oncohematological pediatric patients requiring allogenic HCT. As expected and as this study proves, inflammation may indeed play a pivotal role in voriconazole or other drug exposure.  Certainly, the medical field needs many more of this kind of studies.

Regarding the manuscript, it is well organized and very well written. It only needs a minor revision before acceptance:

We thank the reviewer for appreciating our manuscript.

Q1. I was wondering the term "pediatric" could be really accepted considering that the study also included patients between 18 and 20 (<21)...that are basically proper adults;

A1. We thank the reviewer for this comment, allowing us to better clarify this issue. As per hospital protocol, patients aged between 18-20 years who were previously followed in the oncohematological pediatric unit for the management of the specific oncohematological diseases received allogenic hematopoietic stem cell transplantation in the Pediatric Oncohematology Transplant Unit. Specifically, only one patient of age ranging from 18 to 20 years was included in our study.

Q2. page 3 line 107: replace "12h" with "12 h" (space);

A2. We thank the reviewer for this suggestion. We replaced “12h” with “12 h”.

Q3. page 6 line 153: delete the comma between twenty and one (correct: twenty one);

A3. Thank you for this suggestion. We deleted the comma as suggested.

Q4. maybe the quality of the figures could be improved.

A4. Thank you for this suggestion. We improved the quality of figures.

Round 2

Reviewer 2 Report

Comments and Suggestions for Authors

The authors made the necessary changes/additions to the manuscript. The article may be accepted for publication.